**www.cambridge.org/ext**

**Review**

Conservation prioritization; EDGE; evolutionary distinctness; evolutionary history; extinction; phylogenetics

**Corresponding author:**
Marcel Cardillo;
Email: marcel.cardillo@anu.edu.au

# Phylogenetic diversity in conservation: A brief history, critical overview, and challenges to progress

## Marcel Cardillo [ID]

Macroevolution and Macroecology Group, Research School of Biology, Australian National University, Canberra, ACT, Australia

## Abstract

Species that are evolutionarily distinct have long been valued for their unique and irreplaceable contribution to biodiversity. About 30 years ago, this idea was extended to the concept of phylogenetic diversity (PD): a quantitative, continuous-scale index of conservation value for a set of species, calculated by summing the phylogenetic branch lengths that connect them. This way of capturing evolutionary history has opened new opportunities for analysis, and has therefore generated a huge academic literature, but to date has had only limited impact on conservation practice or policy. In this review, I present a brief historical overview of PD research. I then examine the empirical evidence for the primary rationale of PD that it is the best proxy for "feature diversity," which includes both known and unknown phenotypic characters, contributing to utilitarian value, ecosystem function, future resilience, and evolutionary potential. Surprisingly, it is only relatively recently that this rationale has been subject to systematic empirical scrutiny, and to date, there are mixed results on the connection between PD and phenotypic diversity. Finally, I examine the least well-studied, but potentially greatest challenge for PD: its dependence on the reliability of phylogenetic inference itself. The very few studies that have investigated this so far show that the ranking of species assemblages by their PD values can vary substantially under alternative, routine, phylogenetic methods and assumptions. If PD is to become more widely adopted into conservation decision-making, it will be important to better understand the conditions under which it performs well, and those under which it performs poorly.

## Impact statement

The concept of quantifying evolutionary history of assemblages of species, as a way of assessing the biodiversity value of different areas, has been advocated for the past 30 years. A large academic literature has developed, that applies evolutionary history (most frequently phylogenetic diversity, or PD) in a variety of ways to conservation problems. However, very little of this literature has examined PD from a critical perspective, and there is mixed evidence about whether PD reliably represents the biodiversity qualities that we expect it to. This review aims to summarize recent research that has begun to examine the rationale for PD empirically, and highlight the challenges that will need to be overcome for PD to become more widely adopted into conservation practice.

## Introduction

Species are the primary currency of conservation, despite the inconsistency of species concepts, the ambiguity and taxonomic instability of many species, or the lack of a strong theoretical justification (Maclaurin and Sterelny, 2008). We mourn the loss of a species far more than the population decline that precedes it. The death of the last known individual of a species is considered a mark of humanity's failure to safeguard biodiversity, even if that species had long since ceased playing a functional role in its ecosystem. Yet species are not all considered equal contributors to what we value about biodiversity. Large species tend to be more highly valued than small ones, mammals more than other vertebrates, primates and carnivores more than rodents (R.M. May noted, "As we move from the furries and featheries, down through the innumerable species of insects, and on down to bacteria and viruses, sentimental concern does not merely wane. It changes sign."). And species that are evolutionarily distinct, with few close living relatives, are often regarded as more worthy of protection than those that are less evolutionarily distinct, with many close relatives to which they are genetically and phenotypically similar.

Some species are known to be the only representative of a higher taxon, a Family or even an Order, perhaps one that was more diverse in past ages (Figure 1). The ancestral lineage of these

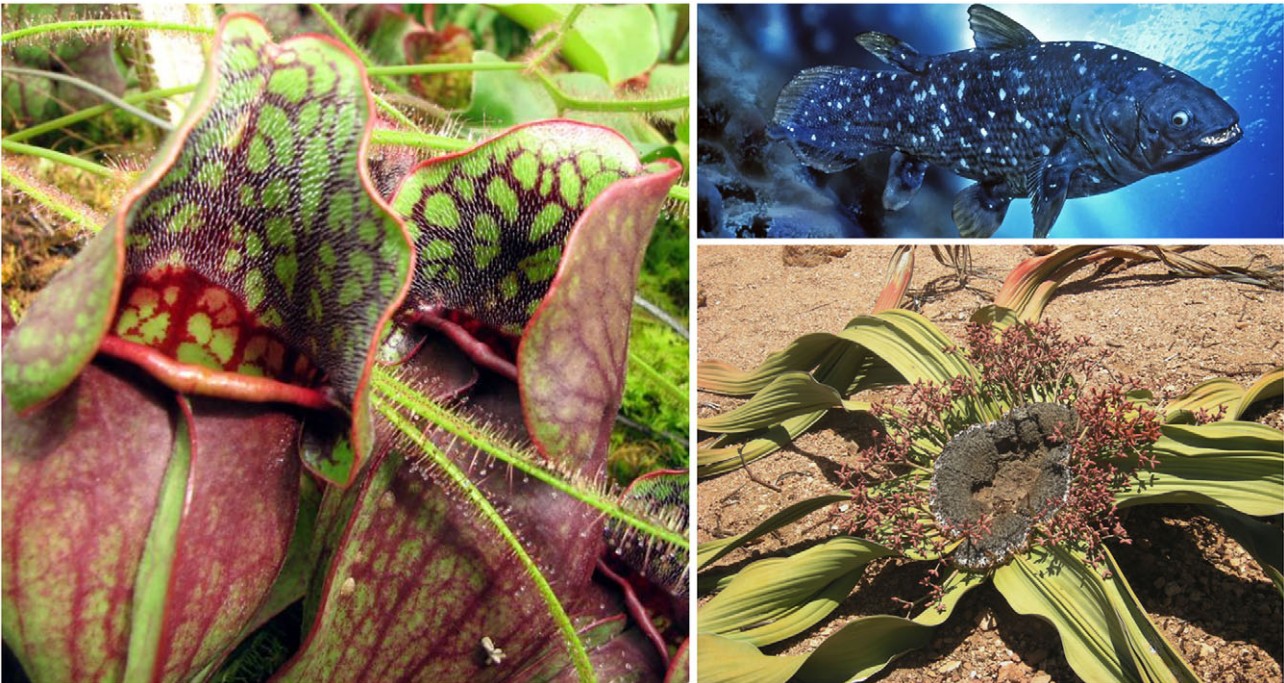

**Figure 1.** Evolutionarily distinct species have long been valued for their unique and irreplaceable contribution to biodiversity. Left: Albany Pitcher Plant, *Cephalotus follicularis* (*source:* https://www.flickr.com/photos/7326810@N08/1555479035/in/photostream/ Attribution 2.0 Generic CC BY 2.0); Top right: African Coelacanth, *Latimeria chalumnae* (*source:* https://www.the-scientist.com/news-opinion/african-coelacanths-may-live-to-be-100-study-68911); Bottom right: *Welwitschia mirabilis* (*source:* https://commons.wikimedia.org/wiki/File:Welwitschia_mirabilis,_m%C3%A4nnl._Bl%C3%BCte,_Namibw%C3%BCste,Namibia.jpg Attribution-ShareAlike 3.0 Unported CC BY-SA 3.0).

species may have diverged from that of their closest living relatives tens or hundreds of millions of years ago. The Albany Pitcher Plant (*Cephalotus follicularis*), for example, is the only species of its family (Cephalotaceae) and represents an origin of plant carnivory that is independent of that in other plant lineages from which it diverged in the mid-Cretaceous Period, over 100 million years ago. The Ganges and Indus River Dolphins (*Platanista gangetica* and *P. minor*) are the only species remaining in the once-diverse and geographically widespread family Platanistidae, of which six extinct genera are known from Miocene fossils. More widely known examples of evolutionarily distinct species include *Welwitschia mirabilis*, a gymnosperm endemic to the Namib Desert and the only species in the Order Welwitschiales; and the Coelacanths (*Latimeria*), the two surviving species of lobe-finned fish in the Order Actinistia.

The extinction of one of these evolutionarily distinct species would represent the loss of a unique and irreplaceable contribution to biodiversity. For this reason, species such as these have long been singled out for special conservation attention. In the past three decades, however, the traditional qualitative value afforded to particular evolutionarily distinct species has given rise to a more quantitative conception of evolutionary history in the context of conservation. This is expressed most commonly in the concept of "phylogenetic diversity" (PD). Essentially, PD amounts to the use of phylogenetic branch lengths as a continuous-scale index of conservation value for a set of taxa. The focus of this review is on PD, rather than single-species measures of evolutionary distinctness. This is because the extension of evolutionary history value from individual species to assemblages brought with it new assumptions and a new kind of rationale that may be less intuitively obvious to many people compared to the rationale for protecting individual species. The theoretical justification for PD and its claims to

primacy as the currency of conservation have been given detailed treatment in recent years by philosophers (Maclaurin and Sterelny, 2008; Lean and Maclaurin, 2016), so I will touch only briefly on this aspect to provide context and background. I will not attempt to review exhaustively the extensive empirical literature on PD, which has been methodically summarized by Tucker et al. (2019). Rather, after presenting a brief historical overview of PD research, I will focus on several questions that help to clarify the current position of PD within conservation biology and are important for the future of PD in conservation practice. How much empirical support is there for the claims made by the advocates of PD in conservation? In particular, I ask if (1) PD serves as a reliable indicator of phenotypic diversity; and (2) PD can be quantified consistently or reliably in the face of the variability, uncertainty, and arbitrary choices that characterize methods for estimating phylogenetic branch lengths.

### Origins of the PD concept

Two seminal papers in the early 1990s mark the onset of a rapid rise in popularity of PD among conservation researchers. Among the first to suggest that phylogeny can be used to avoid treating all species as equal in measures of diversity were (Vane-Wright et al., 1991). Their measure of "taxonomic distinctness" was based on the cladistic information content of a cladogram: that is, the number of monophyletic groups (clades) in which each taxon can be placed, with a higher value for taxa with a more limited clade membership portfolio. In many phylogenetic trees, such taxa often belong to "basal" or early branching lineages. Vane-Wright et al. (1991) showed how taxonomic distinctness weights might be used in complementarity-based algorithms to select priority areas for conservation that maximize the amount of evolutionary history represented by a given number of taxa protected within reserves.

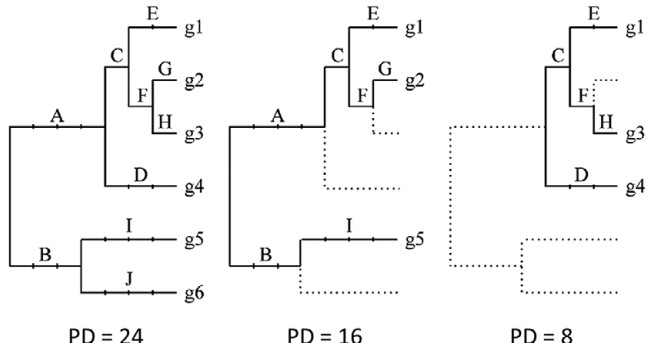

**Figure 2.** Calculation of Faith's phylogenetic diversity (PD) for a set of species g1–g6. PD is the sum of the branch lengths (indicated here by the division of branches into intervals) along the shortest paths connecting the set of species. The panel in the center shows how a small subset of species can capture a large proportion of the evolutionary history of the whole group (left panel), if the paths connecting them traverse the root of the phylogeny. In the panel on the right, a different set of three species has a much lower PD score because they are more closely related to one another. *Source:* Rodrigues and Gaston (2002).

Soon after, Faith (1992) presented PD: a measure of diversity in which taxa are weighted by the number of character-state changes along branches of a phylogeny. The PD for a set of taxa is simply the sum of the number of character changes (or the branch lengths) in the "minimum spanning tree" formed by those taxa and the phylogenetic branches that connect them (Figure 2). The key difference between PD and Vane-Wright et al.'s taxonomic diversity was that the amount of evolutionary change along branches, including the evolution of convergent characters (those that arose independently on several lineages) contributes to the PD value. For taxonomic diversity, the only characters that determine the value of the metric (implicitly) are the shared derived characters that define monophyletic clades. A number of alternative ways of quantifying evolutionary history, both for individual and multiple taxa, have been proposed over the years (e.g., Redding et al., 2008; Chao et al., 2010; Faith, 2016), but Faith's original PD measure is the one that has gained the most traction. Of course, in the years since the original PD paper was published, genomic data has taken over from phenotypic characters as the primary basis for inferring phylogeny, but PD can be calculated equally well from molecular branch lengths, whether in units of evolutionary change or of time.

## The rationale for PD

It has been argued that evolutionarily distinct species (and biodiversity more generally) have intrinsic value; that is, a value in and of themselves that is independent of any rational agent doing the valuing, but this justification for biodiversity conservation has philosophical difficulties (Maclaurin and Sterelny, 2008; Lean and Maclaurin, 2016). In my view, the traditional regard for evolutionarily distinct species as being worth conserving is closer to what McNeely et al. (1990) called "existence value": many people are happy knowing that such species exist, and would feel some kind of loss on an emotional level if they went extinct. The rationale for PD, on the other hand, has always tended to emphasize the instrumental, or utilitarian, values of biodiversity. From the outset, PD was intended to be an indicator of "feature diversity": the diversity of phenotypic characters (both known and unknown) represented by a set of species, which is the real target of conservation (Faith, 1992).

Why is it considered desirable to conserve feature diversity? There are two basic arguments. The first is that maintaining enough phenotypic or ecological diversity allows ecological communities and ecosystems to continue to function and cope with changing and uncertain future conditions (Faith, 1992, 2016). A similar idea applied to intraspecific populations is that conserving genetic diversity, in particular adaptive genetic variation, is a means of maintaining adaptive evolutionary potential, which is regarded as critical for the resilience of populations to rapid environmental change (Moritz, 2002; Sgrò et al., 2011). The second argument for conserving feature diversity is that it maximizes the utilitarian values of biodiversity, economic or otherwise. This includes values that can be "cashed in" immediately (demand value), but it also includes those values that are yet to be realized, or may not yet have been discovered. A frequent example provided for the latter is the future financial value of pharmacologically useful compounds that are likely to be discovered through bioprospecting (Crozier, 1997). Under both of these arguments, conserving maximum feature diversity increases "option value": this is the additional value placed on a sample of biodiversity, over and above its immediate demand value, that derives from having the option of reaping future benefit left open (Maclaurin and Sterelny, 2008; Lean and Maclaurin, 2016).

## The popularity of PD in academic studies

Academic interest in PD has flourished since the early 1990s, and it is worth a brief examination of the reasons for this. Firstly, perhaps, the rationale is compelling: there is an intuitive logic to the idea that if we have to choose between species to conserve (which we do, given the ubiquitous shortfall in conservation funding), then all else being equal, the ones that capture more evolutionary history ought be prioritized. Secondly, the concept itself is simple and quite elegant. The idea of summing branch lengths to represent the amount of evolutionary history captured by a set of species is just as intuitively easy to grasp as many other measures of diversity. PD is computationally undemanding to calculate, provided a representation of phylogeny is already available. The growing popularity of the R language has brought about an increased fluency in handling and analyzing phylogenetic data among ecology and conservation researchers. Tools provided in R packages such as APE and Picante have made it a simple matter to calculate PD and incorporate it into a wide range of analyses. The timing was probably also important: PD was introduced at a time when all things phylogenetic were being adopted into the mainstream of evolution and ecology research, driven by interest in phylogenetically informed comparative methods (Harvey and Pagel, 1991), new macroevolutionary models (Nee et al., 1992), and the integration of phylogeny into community ecology (Brooks et al., 1991; Haydon et al., 1993). During the 1990s, the publication of molecular phylogenies increased rapidly, providing abundant new raw material for the calculation of PD and exploration of its patterns and nuances.

## An overview of analyses of PD patterns

In the first decade after the introduction of PD, a major focus of interest was the effect of species extinctions on the loss of evolutionary history. Many of these studies focused on clade-level patterns (either hypothetical or real clades), rather than spatial patterns. One of the earliest of these studies showed that for a given proportion of species lost, a smaller proportion of the clade's total branch length is lost, so that conserving a modest proportion of species should, in principle, protect much of a clade's evolutionary history (Nee and May, 1997). However, the relative amount of

evolutionary history lost when species go extinct increases when there is substantial imbalance in the phylogeny, the legacy of heterogeneous speciation and extinction rates (Nee and May, 1997; Purvis et al., 2000; von Euler, 2001; Vamosi and Wilson, 2008). The amount of evolutionary history lost increases further still when species extinctions are phylogenetically nonrandom – which is a reasonable expectation, given that currently threatened species are more likely to be found in some higher taxa than others (Purvis et al., 2000, von Euler, 2001, Vamosi and Wilson, 2008). All of these studies were couched in terms of conservation priorities, although none aimed to make specific or direct recommendations for conservation planning and management. They are what Cardillo and Meijaard (2012) termed "call to arms" studies that aimed primarily to draw attention to the consequences of the extinction crisis for the erosion of biodiversity at large scales.

These clade-level patterns were soon extended to spatial analyses of geographic assemblages, which aimed to connect PD more closely with practical conservation decision-making, in the way originally envisaged by Vane-Wright et al. (1991) and Faith (1992). At large geographic scales, spatial patterns of PD are shaped by the history of speciation and extinction in different regions (Davies et al., 2008; Warren et al., 2014). While measures of evolutionary history are generally correlated with species richness, the two do not always align closely and regions of highest aggregate evolutionary history can be very different from regions of highest species richness (Safi et al., 2011; Fritz et al., 2012; Honorio Coronado et al., 2015; Voskamp et al., 2017; Rapacciuolo et al., 2019; Hu et al., 2021). A common practice has been to map the component of variation in an evolutionary history measure that is independent of species richness, for example by using regression residuals or values that are standardized against a null model (Davies et al., 2008; Safi et al., 2011; Fritz et al., 2012; Honorio Coronado et al., 2015; Carvalho et al., 2017; Voskamp et al., 2017; Rapacciuolo et al., 2019; Gumbs et al., 2020; Hu et al., 2021). However, the spatial patterns that result from this can be hard to interpret, because they are often heterogeneous or fragmented, or do not correspond in an obvious way to major climatic or biogeographic zones. It is also difficult to know what such patterns mean from a conservation perspective. Should regions of high PD *relative to species richness* be afforded higher value than regions of high total PD?

Some of the attempts to use large-scale spatial patterns of PD or other evolutionary history measures to guide conservation decisions have taken a gap analysis approach, by asking whether the existing network of protected areas adequately captures PD for a given taxon. In this way, gaps in the protected area network can be identified and recommended as priority areas for future expansion of the network, although there is rarely any objective way of deciding on an "adequate" level of coverage. At a global scale, these kinds of studies have revealed that a high proportion of the priority conservation areas for terrestrial vertebrate and angiosperm PD are unrepresented within current protected areas (Daru et al., 2019; Robuchon et al., 2021). At regional scales, the results have been more case-specific, with some studies finding that PD is relatively well-represented within protected areas (Quan et al., 2018; Aguilar-Tomasini et al., 2021; Llorente-Culebras et al., 2021), and others finding that it is not (McCarthy and Pollock, 2016; Franke et al., 2020; Oliveira et al., 2021).

When the configuration of priority conservation areas needs to be optimized in the face of cost constraints, the "hotspot" approach of simply skimming off areas of highest diversity for protection is not a very efficient way to meet predefined conservation targets

(Balmford and Gaston, 1999; Margules and Pressey, 2000). For this reason, the past 20 years have seen the development of a substantial literature on the use of complementarity-based algorithms (which focus on marginal gains rather than hotspots) to select priority areas for PD conservation. This application of evolutionary history measures was discussed by Vane-Wright et al. (1991) and Faith (1992), but it took some years before both phylogenetic data and computing power were sufficient to solve optimization problems involving PD (Rodrigues and Gaston, 2002). The question most frequently asked by these studies is how closely priority areas selected using taxonomic richness criteria coincide with those selected using evolutionary history. Again, the answers seem to be mixed and case-specific. Some studies have found that species or higher-taxon richness are a good surrogate for PD and there is a high degree of overlap in priority areas selected under both criteria (Rodrigues and Gaston, 2002; Strecker et al., 2011; Pollock et al., 2015; Pollock et al., 2017; Rapacciuolo et al., 2019). Others have found that richness-based and PD-based priority areas differ substantially, so that a focus on protecting maximum taxonomic richness does not adequately protect evolutionary history (Forest et al., 2007; Pio et al., 2011; Carvalho et al., 2017).

## How well does PD predict feature diversity and biodiversity values?

The vast majority of studies involving PD have focused on analyzing patterns of PD, rather than testing the basic assumptions that PD represents feature diversity, and that feature diversity represents utilitarian value, ecosystem function, or evolutionary resilience and potential. It is only fairly recently that some studies have begun to take a critical look at the empirical support for these assumptions (e.g., Kelly et al., 2014; Mazel et al., 2017; Tucker et al., 2018, 2019). One difficulty with evaluating the results of these analyses is that in the PD literature, there has been little clarity or consistency in the definition of feature diversity. Many studies seem to use the terms "feature diversity" and "functional diversity" (FD) synonymously, but Owen et al. (2019) caution against this, arguing that the two concepts are distinct and should not be conflated. In their view, feature diversity encompasses all of the diversity of traits – both known and unknown – possessed by a set of species, while FD is a more narrowly defined subset of feature diversity, based only on traits that have ecological function. For this reason, Owen et al. argue that recent tests of relationships between PD and FD (e.g., Mazel et al.,2018a) do not, in fact, test the ability of PD to act as a proxy for feature diversity. This raises the question of whether feature diversity, defined in this way, can be regarded as a scientifically tractable concept. It means that any empirical test of the relationship between PD and any aspect of phenotypic diversity potentially could be dismissed as not having adequately tested the concept of feature diversity, which by definition is unable to be quantified.

On the other hand, Tucker et al. (2019) describe a more tractable conception of phenotypic diversity that prevails in the ecological literature, as the range of values of any *measurable* trait. Tucker et al. (2018) adopt a pragmatic working definition of phenotypic diversity as "the variation in all ecological or functional traits, which includes a wide variety of physiological, phenological, morphological, and behavioral measures," which they refer to as "FD" to align with the general use of this term in the ecological literature. The advantages of this definition are that phenotypic diversity is measurable and thus testable, and expected associations between phylogenetic and phenotypic diversity can be derived from

macroevolutionary models. In the following discussion, I will follow Tucker et al and use "FD" to represent all conceptions of phenotypic diversity that commonly appear in the PD literature, while acknowledging that these may not properly represent feature diversity in its strict sense.

Studies of the links between PD and FD can be divided into two basic kinds: those that deal with phylogeny-based patterns, and those that deal with area-based patterns. The expectation that PD and FD should be correlated on a phylogeny follows from basic models of trait evolution. The simplest evolutionary model for continuous traits is Brownian Motion, which describes independent, random, nondirectional and unbounded drift in trait values along the branches of a phylogeny. The degree to which the variance in trait values among the tips of a phylogeny, given the lengths of the branches separating them, is consistent with a Brownian Motion model is known as phylogenetic signal. Many species traits (ecological, morphological, physiological, or otherwise) show phylogenetic signal, although it varies in strength among traits and among taxa (Freckleton et al., 2002; Blomberg et al., 2003). This fact suggests that phylogenetic distance among taxa should often correlate positively with trait distance.

However, this does not necessarily mean that PD will usually be a better predictor of FD than species richness, or that a subset of taxa chosen from a phylogeny to maximize PD will also maximize FD. The processes and history of trait evolution can be complex, and Brownian Motion is not always the model most consistent with the data. A trait may be subject to fitness constraints on extreme values; the speed of trait evolution may have varied through time, as a result of adaptive radiation or changing selective regimes; and the speed of trait evolution may have varied among lineages across different parts of the phylogeny. The different processes and patterns of trait evolution affect the PD–FD relationship. The correlation is predicted to be weaker, for example, when traits have evolved under an early burst model (faster evolutionary rates earlier in a clade's diversification) compared to a Brownian Motion model, in which rates are homogeneous through time (Tucker et al., 2018). Simulations and meta-analysis of real datasets have both revealed that positive correlations between phylogenetic and functional distance tend to be restricted to relatively short distances on phylogenies, an effect that becomes more marked as the degree of homoplasy increases (Kelly et al., 2014). This means that as sets of taxa are expanded to include increasingly distant relatives, the capacity of PD to serve as a proxy for FD declines. Furthermore, because a subset of species that maximizes PD is usually distributed nonrandomly on the phylogeny, it can be possible for the maximum-PD set to be a worse predictor of FD than a random set of species (Mazel et al., 2017).

Perhaps more relevant to conservation planning are area-based patterns: correlations between PD and FD, or congruence between the maximum PD set and the maximum FD set, across geographically defined assemblages. In many case studies, there is a high degree of correlation between PD and measures of FD across assemblages, but because both are highly correlated with species richness it is difficult to infer a strong, direct association. This was demonstrated explicitly in a study of the spatial distribution of PD, FD and species richness of plant assemblages in the Pyrenees (Pardo et al., 2017). In this study, there is a high degree of correlation among these three facets of diversity, with PD and species richness equally strongly associated with FD. However, the association between PD and FD largely disappears when the co-association with species richness is accounted for by calculating richness-independent measures of PD and FD.

On the other hand, results of some studies suggest that PD can serve as a useful proxy for diversity of phenotypic traits, including traits with utilitarian value. For the flora of South Africa's Cape region, Forest et al. (2007) showed that not only does prioritizing PD lead to different conservation decisions than prioritizing taxon richness, but PD does a better job of predicting the distribution of plants with economic or medicinal utility. On a global scale, Molina-Venegas et al. (2021) also found that PD captures a greater range of recorded economic values associated with plant taxa, compared to randomly selected taxa. This was not a spatial analysis but was based on selecting taxa from global or continental phylogenies, which seems more consistent with conservation decision-making that emphasizes species rather than areas.

Perhaps the most compelling support for area-based conservation of maximum PD comes from studies of the links between PD, feature diversity, and ecosystem function. The response variables in these studies are the emergent properties of an ecosystem (such as biomass production), so by definition the sets of taxa chosen must be not just spatially congruent, but functionally part of the same ecosystem. The link between PD and ecosystem function is based on a complementarity effect: species with low niche overlap will access different resources and compete little, so that their combined performance when they co-occur (hence ecosystem function) should be greater than that of species with more overlapping resource use. Therefore, if PD predicts complementarity in resource use, it should also predict ecosystem function (Cadotte et al., 2008; Cadotte, 2013). A number of studies of experimental plant communities have found that PD does indeed do a better job than species richness or FD of predicting biomass production (Cadotte et al., 2008; Flynn et al., 2011; Cadotte, 2013). The scale and scope of such studies are necessarily limited by the need to measure ecosystem function in a controlled experimental situation, and it is unclear whether the ecosystem function rationale for PD could be tested for larger or more phylogenetically disparate assemblages across large geographic regions.

## The elephant in the room: Reliability and uncertainty of phylogenetic inference

The final part of this review will focus on the aspect of PD that has received the least attention, but may turn out to be the most important: the dependence of PD measures on the reliability of phylogenetic inference itself. A phylogenetic tree is not really a "reconstruction" of the true, unobservable evolutionary history of a clade, but a hypothesis or inference, based on our assumptions about how evolution led to the data we can observe (Baum and Smith, 2013; Bromham, 2019). Inferring phylogeny requires a large number of methodological decisions, some of which are theoretically well-supported, while others are chosen purely for tractability. The results of a phylogenetic analysis, including branching relationships among taxa, node ages, branch lengths, and measures of support or confidence, are sensitive to these decisions and assumptions, and to the suitability, quality, and completeness of the data. How all of these considerations affect phylogenetic inference is of course a huge field of research, but one that has, until very recently, failed to penetrate the PD literature. Although the potential sensitivity of PD values to the uncertainty of phylogenetic inference was recognized from the beginning (Faith, 1992; Crozier, 1997), there has been very little systematic investigation of this in the three decades since.

Bromham (2019) describes some of the ways in which assumptions and methodological decisions can lead to uncertainty or

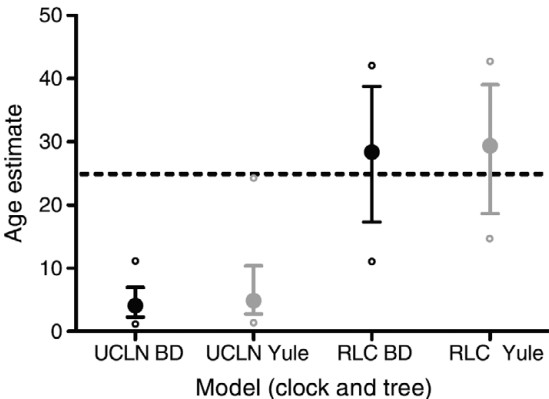

**Figure 3.** Different specifications of models underlying molecular date estimates can lead to very different outcomes. In these simulated clades, alternative combinations of models of diversification through time (Yule vs. Birth-Death) and variation in substitution rates across branches (Uncorrelated Lognormal vs. Relaxed Local Clock) generate crown age estimates that vary from 3 to 30 million years. *Source:* Crisp et al. (2014).

systematic bias in molecular date estimation. These include assumptions about the homology of sites in an alignment of DNA sequences; adequacy of substitution models to account for the data (as opposed to relative goodness of fit); whether variation in substitution rates across branches or over time can be adequately modeled; completeness and randomness of sampling among lineages; constancy of diversification rates; and the ability of new data to change prior beliefs in a Bayesian analysis. Violation of all of these assumptions is routine and can alter estimates of divergence times substantially. In some cases, divergence times have been shown to vary up to seven or eightfold across different fossil calibration scenarios (Sauquet et al., 2012), or under different molecular clock models (Crisp et al., 2014; Figure 3).

From the point of view of PD, the important question is whether the degree of uncertainty that is typical in phylogenetic inference leads to unacceptably low consistency and reliability in PD estimates, and in conservation prioritization analyses that use PD. Few studies have explored this question so far. Park et al. (2018) showed that the common practice of constructing "community phylogenies" (i.e., a phylogeny including only members of geographically-defined assemblages rather than complete clades) leads to underestimation of PD because of under-sampling of the clades from which community members are drawn. Not surprisingly, this effect is more severe when ecological filtering processes (such as competition between close relatives) cause communities to be overdispersed, that is, where species are less closely related than expected from null models. Importantly for PD-based spatial prioritization, this does not simply lower PD values consistently across communities, but alters the rank order of communities by up to 50% when they are ranked by PD values. Ritchie et al. (2021) used a similar approach to investigate the sensitivity of PD rankings to assumptions about variation in evolutionary rates between and along branches, and about timescale calibration methods. Again, it was found that PD values calculated from inferred phylogenies were prone to error (average of 6–14%, and up to 23–38%, difference from true, simulated phylogenies), that the degree of error varied depending on the assumptions, and that out of 100 simulated assemblages ranked by PD, 8–9 were incorrectly included or excluded from the top 10 positions. Ritchie et al. note that when it comes to conservation decision-making it is important to characterize the risk arising from possible worst-case scenarios, so that the

maximum error values they report may be more consequential for conservation than the mean values.

Another consideration is whether the phylogeny used for PD calculations is a phylogram (branch lengths in units of evolutionary change) or a chronogram (branch lengths calibrated to an absolute timescale). Both are commonly used to calculate PD (Elliott et al., 2018), although the phylogram is more in keeping with the original aim of PD to capture feature diversity, while the chronogram is perhaps more appropriate for questions about evolutionary history as something valuable for its own sake. Elliott et al. (2018) have shown that the same data used to construct phylograms and chronograms can lead to very different spatial patterns of PD in Australian and New Zealand plant groups: in some cases, PD hotspots occupy entirely different parts of the Australian continent depending on the type of phylogeny used.

Although investigations of the sensitivity of PD to phylogenetic choices and uncertainty have only just begun, the early signs are that alternative, equally well-justified assumptions and methodological decisions about phylogenetic inference can lead to very different spatial patterns of PD and conservation priorities. However, it is still difficult to know how general these results are, and whether there are particular, easily identified conditions under which the uncertainty and variability in PD values can be limited to acceptable levels. Another thing that remains unexplored so far is the effect that variability in spatial patterns has on the outcomes of algorithmic conservation planning or reserve-selection analyses.

## Conclusion: Where to for PD?

PD has received an enormous amount of academic attention for the past three decades: Faith's original PD paper has been cited over 4,500 times. Yet the adoption of PD into conservation practice (including policy, legislation, planning, and management, at international, national, and subnational levels) appears still to be very limited. The most frequently cited example of a practical application of evolutionary history more generally is the Zoological Society of London's EDGE (Evolutionarily Distinct, Globally Endangered) program (Isaac et al., 2007). This program has had much success in raising public awareness of threatened species, including many that are not widely known, by highlighting the ones that are also evolutionarily distinct. EDGE or evolutionarily distinct species have been recognized as a high priority under IUCN Resolution WCC-2012-Res-019-EN (IUCN, 2012) and by the IUCN Species Survival Commission (https://www.iucn.org/our-union/commissions/species-survival-commission/partners-and-donors/ssc-edge-internal-grant), and some international charities such as On the Edge (https://www.ontheedge.org) have EDGE species as their focus. But although they both have origins in the concept of evolutionary distinctness, EDGE differs from PD in a way that is fundamentally important, because it is not a measure of diversity: it shifts the focus of conservation attention away from assemblages of species and back to the old idea of valuing individual species for their uniqueness. However, EDGE differs from the traditional, qualitative value placed on evolutionarily distinct species in assigning species a numeric score to indicate their value. EDGE also differs from PD in that it is only partly a measure of evolutionary distinctness, with threat status and evolutionary distinctness contributing equally to the score for each species.

Perhaps due to the wide success of EDGE in drawing attention to evolutionary distinctness, PD does appear to be making some inroads into international policy. The most prominent examples

of this are the adoption of PD as an indicator under the Nature's Contributions to People category of the IPBES Global Assessment on Biodiversity and Ecosystem Services (Brauman et al., 2020), and the listing of PD as one of a number of Complementary Indicators of progress toward two goals under the draft Global Biodiversity Framework (Convention on Biological Diversity, 2022). International agreements such as these provide an increasingly important global context and framework that can shape conservation policy at the national level, and it may be that PD begins to find its way into conservation practice in some countries. As yet, however, there is little evidence that this has happened, and it cannot yet be said that PD is a prominent part of the prevailing conservation paradigm.

If PD is to be adopted into the mainstream of conservation practice, it will be all the more important to fully understand the connections between PD and the biodiversity qualities that we want it to represent. Assuming that PD always does an adequate job of representing these qualities could lead to poor outcomes for conservation, so we need to know when it does do an acceptable job and when it does not. Likewise, it is unwise to assume that phylogenetic branch lengths represent evolutionary history precisely and accurately. A better understanding of how phylogenetic error and uncertainty affect conservation decision-making involving PD is one of the most pressing current issues that needs to be addressed with further research. More than anything else, this particular issue highlights a key problem for the wider adoption of PD: considerable specialist expertise is required for a critical understanding of what phylogenetic branch lengths represent, and of the data, models, and methods of inference from which they are derived. The policymakers who decide on metrics and indicators for biodiversity goals do not necessarily have this expertise, and it is critical that the scientific advice provided to them on the utility and limitations of phylogenetic information is balanced, realistic, and free from advocacy.

There are two other important challenges to overcome in order for PD to become more widely applied. One of these is simply data availability: despite the massive growth in generation of molecular data and publication of phylogenetic trees, the majority of the world's biodiversity is still unrepresented in genomic databases, and most of the tree of life remains unknown. This is especially true for some of the world's most biodiverse taxa, including arthropods and angiosperms. Although there are ambitious plans to sequence all described eukaryotic species, estimating PD for many assemblages will probably continue to be based on relationships between higher-level taxa or incomplete data for many years to come, and the possible limitations this imposes on PD estimates needs to be generally understood. While some degree of data incompleteness can be overcome using imputation methods (Gumbs et al., 2018), for poorly known taxa unrealistic data requirements may make it difficult for PD to compete with simple species richness as the primary, basic currency of conservation.

Finally, those of us who are familiar with PD as an academic concept should not assume that the practical conservation relevance of phylogenetic branch lengths is necessarily obvious to most conservation managers, policymakers, funding agencies, government departments, or the general public. How meaningful is the difference between 20 million years and 25 million years of evolutionary history? Do option value or "evolutionary potential" over the next few million years rate highly as priorities for many people, in the way that preventing the extinction of the Siberian tiger or Ganges River dolphin do? With its species-specific focus, the EDGE program has captured public imagination and continues a long tradition of valuing unique and distinct species. Doing the same for assemblages of species, especially when faced with uncertainty over what is being represented, is perhaps the biggest challenge that will need to be overcome for the conservation of PD.

**Open peer review.** To view the open peer review materials for this article, please visit http://doi.org/10.1017/ext.2023.8.

**Acknowledgments.** I thank Lindell Bromham, Ben Scheele, Alex Slavenko, and Sam Passmore for comments and feedback, and the two peer reviewers for their thoughtful and constructive reviews.

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
