## [Reviewer Report]

*Comments to Author*: In principle, a review of the role of PD in conservation is a welcome piece, however this paper is in practice simply an opinion piece critiquing PD, it cannot be considered a review as it omits multiple important advances and applications that have happened in recent years. As well as general misunderstandings throughout, it also mis-represents the author’s previous work to fit the narrative agenda of framing PD in a negative light. The abstract and conclusion both use highly loaded, inflammatory language, unsupported by the limited evidence presented in the paper. This paper should, at the very least, be significantly revised to include the omitted aspects, as well as in presenting a more balanced view of progress. However, as it stands, the paper also does not seem to be in keeping with the stated aims and scope of the journal, and should also be reframed as such to proceed.

I present the key concerns as follows:

1. The paper claims that PD is not used or of any particular use for conservation, ignoring multiple applications, with a non-exhaustive list of applications as follows:

a) explicit recognition by IUCN of its importance with the establishment of the IUCN SSC Phylogenetic Diversity Task Force www.pdtf.org in 2019, as a consortium of experts aiming to do exactly what the author says is a challenge: “bridging the divide between academic conservation science and the scientific requirements of conservation policymakers and planners” (L381-383).

b) building on the 2012 IUCN Resolution on halting the loss of evolutionarily distinct lineages https://portals.iucn.org/library/sites/library/files/resrecfiles/WCC_2012_RES_19_EN.pdf

and recent call by Diaz et al. 2019 to prioritise the conservation of evolutionarily distinct lineages across the tree of life in the Convention for Biological Diversity’s Global Biodiversity Framework https://www.science.org/doi/abs/10.1126/science.abe1530.

c) PD’s increasing influence on conservation activities of the IUCN SSC, recognised as important in prioritising conservation activities in multiple Specialist Groups e.g. in the goals of the Small Mammal Specialist Group, the Amphibian Specialist Group; and the activities of many others funded through the SSC EDGE Grants https://www.iucn.org/our-union/commissions/species-survival-commission/partners-and-donors/ssc-edge-internal-grant.

d) Dedicated and increasing donor and practitioner support globally for conserving species important to maintaining PD, specifically EDGE species and Zones, by multiple organisations, e.g. ZSL, On the Edge, re:wild, Rainforest Trust among others.

e) the adoption of Phylogenetic Diversity by IPBES as an indicator for multiple aspects of Nature’s Contributions to People in their Global and Regional assessments https://ipbes.net/global-assessment.

f) The inclusion of Phylogenetic Diversity in the draft Global Biodiversity Framework as a Complementary indicator for Goal B, and the paired EDGE Index as a Component Indicator for Goal A, see here for technical submissions https://www.pdtf.org/publications leading to the most recent CBD COP15 draft document listing the indicators: https://www.cbd.int/doc/c/0524/cc9d/99da38b8be1522bd3fd97e43/cop-15-02-en.pdf, full details of the indicators as described in the pre-print https://www.biorxiv.org/content/10.1101/2021.03.03.433783v1.full.

These two indicators were even proposed by some Parties to be considered as Alternative Headline Indicators during the CBD technical meetings in Geneva, March 2022 https://www.cbd.int/doc/c/f191/8db7/17c0a45b42a5a4fcd0bbbb8c/sbstta-24-l-10-en.pdf.

g) The inclusion of Phylogenetic Diversity in the Multi-Dimensional Biodiversity Index developed by UNEP-WCMC and already incorporated by pilot countries, Soto-Navarro et al. 2021 https://www.nature.com/articles/s41893-021-00753-z.

h) Reporting on EDGE species by WDPA’s Protected Planet e.g. https://livereport.protectedplanet.net/pdf/Protected_Planet_Report_2018.pdf.

i) Systematic conservation planning in Australia using PD and associated metrics, e.g, Rosauer et al. 2018 https://conbio.onlinelibrary.wiley.com/doi/10.1111/conl.12438 ; Laity et al. 2015 https://geobon.org/downloads/scientific-publications/2015/1-s2.0-S0048969715300498-main.pdf

Unfortunately, the author does not seem to understand the principles of ZSL’s EDGE of Existence programme as the most widely cited application of PD in conservation, claiming that it is “quite far removed from PD…[shifting] focus back to the old idea of valuing individual species for their uniqueness” (L369-371). The EDGE programme, and the multiple papers presenting EDGE assessments across multiple taxonomic groups, clearly recognise PD as foundational to this work, and it is highlighted by the IUCN SSC PDTF as being a practical application of PD in conservation. It is quite a disjointed and somewhat contradictory narrative for the author to present the conservation of evolutionary history in species as then leading to the quantification of PD but then to distance the consequently developed EDGE metric from PD. 

I would also suggest the author may like to undertake a more rigorous review of the applications of PD in conservation than post a request on twitter https://twitter.com/MarcelCardillo/status/1549624820316286981?t=qsro5ScTF9w_Ko64gc3Kug&s=31.

2. The paper neglects or downplays evidence in support of the application of PD in conservation, to support a negative and unbalanced narrative, in general there is a misunderstanding of the debate, evidence and findings to date, in a variety of ways. 

PD is described by the author as “a continuous-scale index of conservation value for a set of species, calculated by summing the phylogenetic branch lengths that connect them.” L5 & 54. But PD is not an index of conservation value – it is a measure of biodiversity, that informs conservation. Conservation does not necessarily seek to maximise PD, but to conserve PD, an important distinction highlighted in Owen et al. 2019 https://pubmed.ncbi.nlm.nih.gov/30787282/.

No academics, practitioners or expert groups such as the IUCN SSC PDTF lay claim to PD’s “primacy as the currency of conservation” (L57) as the author asserts. This is especially important as conservation does not work this way in practice, with any single prioritisation scheme - in reality, the intention of PD-informed conservation such as the EDGE of Existence programme and other PD-initiatives (as outlined above) is to complement current conservation efforts and prioritisations, and seek to highlight where valuable species and areas may otherwise be overlooked.

The author does not seem to understand conservation practice, claiming “simple species richness as the primary, basic currency of conservation” (L378). For some time now, the literature has recognised that conservation efforts should not be based on species richness alone but on additional metrics, such as species composition, endemism, functional significance, and the severity of threats (and, increasingly, evolutionary distinctiveness). For example biodiversity hotspots are based on endemism, Key Biodiversity Areas on the presence of trigger (threatened) species, the IUCN Red List on extinction risk, and indeed efforts are typically made to control for species richness in spatial or species-based prioritisation analysis.

The author decides throughout that feature diversity equates to all work on PD / functional relationships, despite feature diversity having a specific definition and this being cautioned against repeatedly, e.g. in Owen et al. 2019. 

The inference that feature diversity and option value (and hence PD) is solely a utilitarian value of biodiversity (L122 + L222) is not supported in practice, as it is contrary to the use of PD as an indicator for multiple NCPs by IPBES and in consideration throughout the CBD (see links above), both of which’s descriptions highlight its importance as a mechanism for ensuring intergenerational equity. Future benefits from biodiversity are of course not restricted to only utilitarian use and include benefits derived from non-utilitarian extrinsic values such as cultural, aesthetic, etc as well as intrinsic values. 

The author critiques data limitations around PD analyses (L379), but has omitted efforts to overcome these, such as that presented in Gumbs et al. 2018 https://journals.plos.org/plosone/article?id=10.1371/journal.pone.0194680; note that this area is also advanced extensively in the pre-print https://www.biorxiv.org/content/10.1101/2022.05.17.492313v1.abstract.

There is some mis-representation of previous findings, most notably being the author’s co-authored Ritchie et al. 2021 paper, but also Kelly et al. 2014 and Mazel et al. 2018 which actually do show strong support for the PD and feature / PD and function relationship in tree space and geographic space respectively. The author dismisses Molina-Venegas’s work and ignores the positive overall findings of Mazel et al. 2018; Tucker 2018,2019, and fails to cite Owen et al. 2019’s response to Mazel et al. 2018.

For Ritchie et al. 2021, specific examples are as follows:

The author claims: “Again, it was found that PD values calculated from inferred phylogenies were prone to error (23-38% difference from true, simulated phylogenies)...” L339-340. 

But in Ritchie et al. 2021 the average % error is 6-14% (i.e. an average of 86-94% accuracy), which in fact is very positive, particularly for true PD estimation from reconstructed trees. The values quoted here are actually the max error, and this should be represented correctly. 

The following phrase is also incorrect “…and that the ranked positions of 100 communities differed between true and inferred community PD by an average of 10-11 places” (L341-342).

In fact, these figures are about species ED rankings (not community PD rankings), and the Ritchie et al 2021 paper says, quite positively by comparison: 

"Looking at how the position of each taxon changed when we used reconstructed ED, we found that taxa were mis-ranked by 10-11 positions on average and 20-40 positions at the 95th percentile compared to their rankings based on true ED values (BEAST, Figure 6; NPRS, Figure S4 available as Supplementary Information). Taxa that were top-ranked in the true tree were substantially more likely to be correctly ranked than those that had ED values in the middle of the ranking. An alternative way to interpret this data is to compare the proportion of the top 10, 50 (and so on) ranked species that are correctly identified under estimation. The above results are then equivalent to saying that 83-87% of the top 10 or top 50 species are correctly identified by estimation, whereas about 90% of the top 80 are correctly identified.”

3. Inflammatory language

The abstract is highly editorialised and does not match the content. This is also the case in the conclusion, claiming that PD currently has no impact on conservation decision making after omitting the multiple (and non-exhaustive) list of advances outlined above. 

In particular, extremely loaded language unreflective of the advances that have already been made appears in the following sentences, which should be entirely revised on the basis of the evidence above:

L8 “has had virtually no impact on conservation practice or policy.”

L19-20 “it will be difficult to envisage a major role for PD in conservation policy and real-world decision making.”

L385-386 “…if that is ever to happen.” 

L386-389 “The second will serve as a reality check on the value of PD for conservation…and help to identify the conditions under which PD might be considered to represent whatever it is that we value about biodiversity.” 

Finally, the dramatically increasing interest in PD-informed conservation over recent years, spearheaded by concerted, cohesive and truly collaborative efforts from scientists, practitioners, donors and policy-makers highlighting the need to incorporate PD in conservation (but not as an exclusive goal), would seem to undermine the author’s claim that “ PD is certainly not a prominent part of the prevailing conservation paradigm.” (L373).

---

## [Reviewer Report]

*Comments to Author*: This opinion piece presents an overview of some of the debates and potential problems with approaches based on the concept of phylogenetic diversity. It is an interesting read, while I don’t necessarily agree with all the points that are made here. 

The section exploring why PD is popular in academic studies argue that it is caused by two factors, the ease of compiling PD and the fortuitous rise of molecular systematics in the 1990s coinciding with the introduction of PD in 1992. While that may be somewhat the case, the way it’s presented makes is a bit disingenuous. Putting the conservation aspects aside, PD and associated metric have proven to be useful tools in deciphering biodiversity patterns and exploring the potential processes behind these patterns. 

The ability of PD to predict feature diversity is certainly a topic that has been widely debated in recent years. It would be important to mention that some of the publication cited here use a narrow view of feature diversity, as pointed out for example by Owen et al 2019 (not cited here; https://www.nature.com/articles/s41467-019-08600-8) in the case of Mazel et al 2018a. In some of these papers, functional diversity is equal to feature diversity, which are in fact, as pointed out by the author, two different concepts. The examples reported in lines 275 to 283 are more in line with the concept of feature diversity (i.e. usefulness of plants), which is broader than functional diversity. 

The last section of the manuscript focuses on the uncertainty in the phylogenetic inference itself, which is certainly not an issue for PD alone. Most of the points raised here are valid and I agree that additional research in this field would be necessary, especially studies using rarefication approach of real, near-complete data, rather than simulated data (such as Ritxchie et al 2021), which have value, but might not be capturing all the complexities of phylogenetic inference. The data needed for rarefication analyses are not readily available, but hopefully these will become more common in the future. 

While I would not necessarily advocate that PD is the silver bullet that can provide all the answers we need in conservation science, it is certainly important to capture the evolutionary dimension of biodiversity, an important contributor to the diversity of life on Earth, when planning conservation actions. PD should be seen as representing one of the many components of biodiversity and should be considered in conservation planning where possible. I feel that this review/opinion piece is rather dismissive in that regard, but maybe it’s the way I interpreted it. In any case, it is, of course, an opinion that the author is entitled to have and several interesting points are made here.

Minor points:

L36: Family and order don’t need to be capitalised

L124: I don’t think that option values generally refer to financial value, so this example (i.e. pharmacologically-useful compound) might not be entirely representative of the general concept of option values. 

L143: I don’t think the R package ape has functions to compile PD.

Figure 2: Faith’s definition of PD includes the root. The PD calculation using the tree on the right excludes the root, so this is not strictly Faith’s PD and more what some have called “local PD”. It doesn’t affect the point that the author is attempting to make here, however.

---

## [Editor Report]

*Comments to Author*: Thanks for submitting your review of phylogenetic diversity. I found it interesting to read, and I’ve sought the opinions of two referees. Both referees note some aspects of communication that could be tempered, albeit that they differ in their recommendations and the length of their reviews. And one is very positive.

I did another editorial read to evaluate the comments you have here in the longer review. In particular, I was interested whether the manuscript, in its current form, is written in a suitably objective tone. Having done that, I do have some sympathy for the comments in the longer review, and I think this is something you could address in a revised version of the manuscript.

To illustrate my impressions, I’ve given a few quick examples below. I only give a few examples, but I’d like you to pay attention to phrasing and accuracy throughout the ms. The paper will have a bigger influence on the field if the phrasing is fair and objective, and does not alienate people who have a different view to the one presented here. I’d also like you to address the points of scholarship and accuracy from the longer review, and to respond to each of the referees' comments in a reponse letter, explaing what you did to the ms (also, doing your changes using coloured text).

--“...accepted, almost uncritically” [we don’t know whether people think critically or not, just by reading the text they wrote, strictly-speaking]

--“...adds nothing to...” [’adds nothing‘ is a strong statement from a statistical perspective. Of example, it could ’add something' even if it added a small amount of additional information]

--“...suffers from the same vulnerability to phylogenetic branch-length uncertainty” [this is a quantitative issue that is tackled only qualitatively. There is a question of effect size. For example, many of the samples given concern unequivocally very long brand distinct lineages (e.g. coelacanths). For these, the effect size of the uncertainties will be small].

In summary, I’m happy to consider a raised submission of the manuscript that addresses the referees' comments. Please do use the information they have given in a constructive way.

When you do resubmit, please provide me with a list of suggested referees in your cover letter. I would like to broaden the range of invited referees here, to ensure the best quality of outcome.

---

## [Reviewer Report]

*Comments to Author*: I continue to welcome the principle of an objective review of the role of PD in conservation. However, despite some acknowledgements of the issues raised being made in the response to reviewers, this hasn’t been sufficiently brought through to the manuscript revisions. This version is written more neutrally, and there are sections which present good insights and value, particularly the phylogenetic inference section. However, the overall paper is still somewhat jumbled, and lacks internal consistency. There are two major areas of concern, plus a few other elements that could be better addressed. 

1. Firstly, in general the paper’s approach falls into the trap of conflating research into a biodiversity metric (PD) with conservation that utilises PD along with measures of vulnerability, as is the main approach for conservation e.g. Brooks et al. 2006, and it is not appropriate to solely use the former to question the latter. 

L63 “I ask if 1) PD serves as a reliable indicator of conservation-relevant phenotypic diversity". This is certainly an ambitious question, but the author does not define ‘conservation-relevant phenotypic diversity’ satisfactorily, and I’m uncertain if this is even possible, thus raising the question of how the author can come to a judgement. The review does not seem to adequately answer this question.

L223-226 “Most authors of papers on PD seem to regard feature diversity implicitly as the variety or richness of phenotypic traits of any measurable kind, including physiological, phenological, morphological, and behavioural traits (Tucker et al. 2019), without explicit consideration of whether the traits are of relevance to the goals of PD.” 

It is unclear what is meant here: what exactly are “the goals of PD”? The goal of maximising PD (vs conserving PD, which is different and should be clearly differentiated in this review) is to retain the broad suite of features precisely because we don’t know what will be useful in the future, so how can it be stated (and by whom?) that any one trait is or is not of conservation relevance? However, in terms of conservation strategies, conserving PD, or maximising threatened PD, becomes the objective, and this needs to be clear if assessing the role of PD in conservation.

As part of this concern, although feature diversity is a central concept, within the manuscript feature diversity is repeatedly used interchangeably with functional diversity, which is a fundamental inaccuracy in this paper and has been cautioned against in the literature – functional diversity is only a subset of feature diversity, and this needs to be made clearer throughout. Since most of the review hinges on this point, they should both be separately defined, especially in relation to their differential use in the various studies cited. Otherwise, the review continues to misrepresent PD as a proxy for a selection of functional traits, rather than representing overall feature diversity. Specific examples as follows:

L109-111 “…PD was presented as a proxy for the diversity of unknown characters. In the age of genomic phylogenetics and open data, this is still the primary rationale for PD.”

The rationale is that it is impossible to know and measure all features of all species, and PD indicates that overall diversity of features. This is not the same thing as the diversity of unknown characters that were withheld from the public domain by scientists, or that can shift in meaning as more data become available. 

L229 – “I will not dwell on the issue of definitions, but will use the term feature diversity to represent all conceptions of functional, trait, or phenotypic richness or diversity that appear in the PD literature.” 

Please do dwell on the issue of definitions, because this seems critical to the entire point of the review. For example, feature diversity when the target of PD conservation is often defined as the variety of different features, measured and unmeasured, represented among species or other taxa, and it is widely acknowledged that studying the link between PD and a narrow selection of traits (i.e. functional diversity) does not represent a test of the PD-feature relationship (e.g. ”It is important to recognise that PD-based prioritisation aims to capture the diversity of evolutionary features of species, both measurable and unmeasurable…FD is just one part of this diversity.” - Griffith et al. 2023; and see Owen et al. 2019 and related articles). By conflating studies focusing on functional trait diversity (a la Mazel et al. 2018) with tests of PD-FD relationship (a la Kelly et al. 2014), the author fails to accurately reflect the literature. This could easily be remedied by spending the necessary effort to clearly define the terms used by the author and in the papers referenced, e.g.: 

L257-258 “Furthermore, because a subset of species that maximizes PD is usually distributed non randomly on the phylogeny, it can be possible for the maximum-PD set to be a worse predictor of feature diversity than a random set of species (Mazel et al. 2017).”

Mazel et al 2017 was referring to functional diversity, not feature diversity. The two are not interchangeable. 

L263-265 “This was demonstrated explicitly in a study of the spatial distribution of PD, functional diversity and species richness of plant assemblages in the Pyrenees (Pardo et al. 2017).”

The author is now talking about functional diversity, but it is not clear whether he considers this to be a component of feature diversity, or is using the terms interchangeably. 

2. Secondly, the author continues to provide a contradictory perception of EDGE and the link to PD. They say that “the EDGE approach represents the current primary practical method to apply PD to conservation” in the response to reviewers, and in the paper outline how PD emerged from work on evolutionary distinctiveness, but then still distance EDGE (Evolutionary distinctiveness weighted by extinction risk) from PD when discussing PD’s uptake in conservation. This arbitrary distancing is the opinion of the author, and is in opposition to the original stated aim of EDGE (from Isaac et al. 2007): 

“Here, we define a simple index that measures the contribution made by different species to phylogenetic diversity and show how the index might contribute towards species-based conservation priorities.”

“This paper describes a new method for measuring species' relative contributions to phylogenetic diversity”

“The EDGE approach identifies the species representing most evolutionary history from among those in imminent danger of extinction. Our methods extend the application of PD-based conservation to a wider range of taxa and situations than previous approaches“.

And also contradictory to empirical data (e.g. see Redding and Mooers 2015, PLOS ONE). If the author’s point is that EDGE is a species-focused approach and their personal concern is only with assemblage-based measures, this is not clearly stated in the review and needs to be brought to the fore. 

Whilst being unclear when critiquing assemblage PD and species-based measures, the author thus chooses to exclude elements such as the paired EDGE indicator, a component indicator in the CBD’s GBF (explicitly linked to—and derived from–the PD indicator), from being classed as advances in PD-informed conservation, which is misleading. 

3. Other points

The correction of the misrepresentations of other research is improved, though still ambiguously worded in a way that could be misconstrued, or selectively presenting certain results, e.g. L334-337. Ritchie et al. 2021 provides stimulating and insightful findings, yet only a narrow set of these findings are highlighted in this review. e.g. choosing to highlight areas of relatively weaker performance of PD while ignoring areas of strong performance, with the previously erroneously cited (and positive, from my perspective) ED results now removed entirely from the review.

Regarding the author’s arguments that the researchers in the field are not concerned with the importance of uncertainty and phylogenetic inference and its implications for measuring PD and its link to feature diversity (e.g. L396-399), while the author highlights some areas that are indeed in need of greater interest, there are multiple examples to the contrary for various aspects of phylogenetic uncertainty/error – this is a very active area that is well-recognised in the literature and was even discussed in early literature around the EDGE metric (e.g. Isaac et al. 2007). 

L354 – the author fails to note here the increased exploration of the impact of different phylogenetic hypotheses in phylogenetic-based work and how to incorporate or address uncertainty where possible (e.g. Jetz et al. 2014 Current Biology, Pollock et al 2017 Nature, Stein et al 2018 Nature Ecol Evo, Rabosky et al. 2015 Evolution, Weedop et al. 2019 Animal Conservation). 

L357 – this sentence is good and should be a call to arms to phylogeneticists to tackle these issues to provide more robust PD calculations. 

L42 – now classified as two species, P. gangetica and P. minor

Positively, I do agree with several of the author’s points: around phylogenetic inference and how increased research into the conditions under which PD works best are exciting avenues, and that PD performs variably at capturing sets of functional traits, and including clarity on both points (for the former: people do care and are working on some aspects of it; for the latter: greater clarity on function vs feature diversity as mentioned above) would go a long way to helping this transition from an opinion piece to a review. These positive elements are being overshadowed by the lack of clarity and conflation, and addressing these aspects will solve the issues outlined above.

---

## [Editor Report]

*Comments to Author*: Thanks for resubmitting your manuscript. You will see that one referee has some further comments for you to consider, but that those comments are now more constrained in scope than before. I would welcome a resubmission of the manuscript that considers these points.

---

## [Editor Report]

*Comments to Author*: Thanks for your considered participation in he peer review process for this paper. I appreciate the seriousness with which you have gone abou incorporating suggestions from both referees. I’m satisfied that you have done essentially a complete job with this and am happy for the paper to be published in it’s current form (pending any input from the technical editorial team of the journal). Thanks again, I look forward to seeing the work published.